Prediction of pathological complete response to neoadjuvant chemotherapy for invasive breast cancers based on longitudinal ultrasound and superb microvascular imaging: a single-center retrospective study

Zuo Yanling 1 2
Zhan Yongtao 3
Zhou Jie 3
Xia Haoming 3
Li Tao 2
Zhou Fan 4
Luo Chunyue 2
Zeng Huafeng 2
Li Yingjia lyjia@smu.edu.cn 1
1 Department of Ultrasound, Nanfang Hospital, Southern Medical University , Guangzhou , Guangdong , China
2 Department of Ultrasound, Affiliated Cancer Hospital & Institute of Guangzhou Medical University , Guangzhou , Guangdong , China
3 Department of Breast Surgery, Affiliated Cancer Hospital & Institute of Guangzhou Medical University , Guangzhou , Guangdong , China
4 Department of Radiology, Sun Yat-Sen University Cancer Center , Guangzhou , Guangdong , China
Guan Fanglin
Electronic publication date: 2025 Oct 14
Publication date: 2025
Volume: 13
Electronic Location ID: e20171
Received 2025 Jun 4; Accepted 2025 Sep 11
Copyright: ©2025 Zuo et al.
Copyright year: 2025
Copyright holder: Zuo et al.
License: This is an open access article distributed under the terms of the Creative Commons Attribution License, which permits unrestricted use, distribution, reproduction and adaptation in any medium and for any purpose provided that it is properly attributed. For attribution, the original author(s), title, publication source (PeerJ) and either DOI or URL of the article must be cited.
License URL: https://creativecommons.org/licenses/by/4.0/

Keywords: Breast cancer, Neoadjuvant chemotherapy, Ultrasound, Superb microvascular imaging, Pathological complete response

Funding: Guangdong Provincial Medical Science and Technology Research Foundation Project No. B2023151 This project was supported by Guangdong Provincial Medical Science and Technology Research Foundation Project (No. B2023151, Tao Li). The funders had no role in study design, data collection and analysis, decision to publish, or preparation of the manuscript.

==============================
Purpose

To examine whether dynamic alterations in conventional ultrasound (US) and superb microvascular imaging (SMI) can act as predictors of pathological complete response (pCR) following neoadjuvant chemotherapy (NAC) in breast cancer (BC).

Methods

This single-center, retrospective study included women with invasive BC who underwent NAC between January 2022 and December 2024. The features of conventional US and SMI characteristics of BC were analyzed before NAC and the change (Δ) after two cycles. Multivariate logistic regression analysis (Forward, Wald, α = 0.05) was used to screen factors independently associated with pCR. Area under the receiver operating characteristic curve (AUC) analysis was performed to confirm the predictive effectiveness and evaluate the internal validity through bootstrap resampling. A nomogram was created to graphically represent the predictive power of the various factors for pCR.

Results

Before NAC, the pCR group exhibited significantly higher negative rates for the estrogen receptor (ER) and progesterone receptor (PR). (P < 0.001 and P = 0.005, respectively) and significantly higher positive rates of human epidermal growth factor receptor 2 (HER2) and echogenic rinds (P < 0.001 and P = 0.029, respectively). Additionally, they exhibited significantly shorter largest diameters (LD) and shortest diameters (SD) (P = 0.001 and P = 0.003). After two cycles of NAC, patients who achieved pCR exhibited a significantly higher proportion of monochrome superb microvascular imaging (mSMI) that had not expanded, as well as disappearance of the echogenic rind (P < 0.001 and P = 0.002). Regarding the rate of change in LD, SD, and vascular index (VI), patients in the pCR group showed significantly higher values than those in the non-pCR group (all P < 0.001). The multivariate logistic regression model identified ΔVI (%), ΔSD (%), and SD to have the strongest association with pCR. The overall multivariate model demonstrated the best AUC (0.963), which was significantly higher than that of any single factor. Bootstrap resampling, calibration plots, and decision curve analysis (DCA) all demonstrated strong performance in both discrimination and calibration.

Conclusion

The baseline status of US and SMI, as well as the longitudinal changes, demonstrated good predictive performance for pCR in BC following NAC.

Introduction

Neoadjuvant chemotherapy (NAC) was introduced by Frei (1982) as systemic chemotherapy for localized tumors prior to radical surgery or radiotherapy. Currently, NAC is the standard treatment for breast cancer (BC) patients to eradicate cancer cells and assess tumor response to systemic therapy (Sabeti et al., 2025). Pathological complete response (pCR) is regarded as a surrogate indicator of favorable overall, event-free, and long-term survival in BC (Early Breast Cancer Trialists’ Collaborative Group (EBCTCG), 2018). Early identification of pCR can promptly guide and adjust the treatment plan, thereby avoiding unnecessary surgery and chemotherapy cycles. Therefore, there is an urgent need to develop methods for the early prediction of pCR.

Magnetic resonance imaging (MRI) is widely regarded as the most sensitive imaging modality for assessing NAC in patients with BC (Hayward et al., 2023; Wang et al., 2023). Nevertheless, literature reports (Kim et al., 2023b) indicate that the combined use of Doppler ultrasound (US) and elastography surpasses MRI in detecting residual cancer after NAC for BC. Additionally, MRI is not suitable for patients with metal implants within their bodies or for those experiencing renal insufficiency; however, US is more convenient, cost-effective, radiation-free, and widely accessible. Therefore, it is highly recommended that patients with BC undergo regular re-evaluation of their response to chemotherapy using US after NAC (Shao et al., 2022). The echogenic rind of the breast lesion on grayscale US has a high positive predictive value (PPV) for diagnosing BC (Watanabe et al., 2021). It will be incorporated into the sixth edition lexicon of the American College of Radiology Breast Imaging Reporting and Data System (BI-RADS). Chung et al. (2022) discovered that triple-negative BC with echogenic rinds before NAC was more likely to achieve pCR. However, there were few documented reports on the changes in echogenic rinds before and after NAC. We aimed to explore whether the presence or absence of a specific factor, and its changes following NAC, were associated with pCR.

Angiogenesis plays a pivotal role in BC progression, significantly contributing to disease aggressiveness, resistance to treatment, and unfavorable prognosis (Zhang et al., 2022). Following NAC administration, the intratumoral vascular diameter and blood flow velocity decrease, posing greater challenges for detecting tumor blood supply (Kuo et al., 2008; Kumar et al., 2010). Contrast-enhanced ultrasound (CEUS) offers detailed insights into the microvasculature and hemodynamics (Zhou et al., 2020). However, this technique requires the injection of contrast agents, which not only increases the cost of examinations for patients, but also prolongs the time doctors spend on image analysis. Superb microvascular imaging (SMI) is a novel, feasible microflow imaging technique that eliminates clutter, retains low-velocity signals, and enhances sensitivity to microvasculature within the tumor without the need for a contrast agent (Kurt et al., 2023).

SMI provides two modes of vascular imaging: color superb microvascular imaging (cSMI) and monochrome superb microvascular imaging (mSMI). The cSMI simultaneously displays grayscale and color information. It features dedicated software to evaluate the vascular index (VI), which calculates the ratio between the pixels of the Doppler signal and those of the total lesion. Studies have shown that VI can help increase diagnostic accuracy in differentiating between benign and malignant breast lesions (Chae et al., 2021; Zhang et al., 2022). However, there have been few reports on the role of VI in predicting response to NAC in BC. The mSMI image is displayed side by side with the grayscale image, which reduces the background signal in the region of interest, focusing solely on the vasculature. The images are similar to those of contrast-enhanced ultrasound, but do not require contrast agents. Diao et al. (2020) reported that breast lesions with a larger scope in contrast-enhanced mode than in grayscale mode are more likely to be malignant. This phenomenon is also observable in the mSMI images. To our knowledge, there is limited understanding of its correlation with the response to NAC in BC.

Therefore, the objective of this study was to investigate whether longitudinal changes in various BC lesion parameters are associated with pCR following NAC.

Materials and Methods

Patients

This single-center retrospective study was conducted on 169 biopsy-proven BC patients prior to NAC at the Affiliated Cancer Hospital and Institute of Guangzhou Medical University (Guangdong, China) from January 2022 to December 2024. The inclusion criteria were as follows: (a) biopsy-confirmed invasive BC; (b) US and SMI examinations performed before NAC and after two cycles; (c) completion of a standardized NAC regimen at our institution; and (d) postoperative pathological evaluation of treatment response to NAC. The exclusion criteria were as follows: (a) no surgery performed at our institution (n = 11), (b) presence of distant metastasis (n = 4), (c) no administration of NAC prior to mastectomy for BC (n = 34), (d) absence or substandard quality of US or SMI (n = 8), (e) incomplete NAC cycles (n = 5). and (f) Patients who had undergone breast surgery, chemotherapy, or radiotherapy, as well as those with foreign materials, such as implants, were excluded. Consequently, a total of 107 patients were included in this study. The study design and protocol were approved by the Ethics Committee of the Affiliated Cancer Hospital and Institute of Guangzhou Medical University (No. KY-2025-08; dated February 5, 2025). Written informed consent was obtained from all the patients.

Ultrasound examinations

All examinations of breast lesions in the US were conducted using US equipment (Aplio 300, Canon Medical Systems Corporation, Tokyo, Japan) with an L14-5 MHz high-frequency linear array transducer. First, two orthogonal grayscale images of each lesion were acquired using grayscale US, and they were recorded to observe whether there were echogenic rinds around the lesion and to measure the lesion’s longest (LD) and shortest (SD) diameters, which were mutually perpendicular. Next, the SMI mode was selected and applied with minimal compression to the skin surface, and the color gain was adjusted to visualize the low-velocity flow. First, the cSMI mode was entered. VI was determined automatically by manually delineating the boundary of the breast lesion on the cSMI image with the richest vasculature. This process was repeated at least three times, and the average was calculated. Subsequently, the system switches to the mSMI mode. For smaller lesions, the mSMI mode stores their maximum cross-sectional image; for larger lesions, it captures cross sections that include the lesion margins. This is for subsequent analysis to determine if the lesion in the mSMI image exceeds the size observed on grayscale US. All patients underwent grayscale US and two modes of SMI examination, before and after two cycles of NAC. The changes in LD, SD, and VI of the tumor were calculated as follows: ΔLD(%) = (L0–L2)/L0, ΔSD (%) = (S0–S2)/S0, and ΔVI (%) = (VI0–VI2)/VI0. In this formula, L0 and L2 represent the longest diameters of the lesion before and after two cycles of NAC, respectively. S0 and S2 represent the shortest diameters, and VI0 and VI2 represent the vascular indexes. All US examinations were conducted by two breast radiologists, both possessing over a decade of expertise in breast imaging and one year of experience in SMI. When analyzing the images, a consensus was reached through discussion in the event of disagreement in opinions.

Clinical and pathologic data

All BC cases were pathologically diagnosed using US-guided core needle biopsy before undergoing NAC. Immunohistochemistry (IHC) of the biopsy specimens provided information on the histologic type, estrogen receptor (ER), progesterone receptor (PR), human epidermal growth factor receptor 2 (HER2), and Ki-67. HER2 expression was considered positive when the HER2 score was 3+, and negative when it was 1+ or absent. For cases with a HER2 score of 2+, silver in situ hybridization (SISH) was used to assess HER2 positivity. HER2 expression was considered positive in the presence of HER2 gene amplification, as confirmed by SISH, and negative in the absence of amplification. When multiple malignancies were present, the largest mass identified through grayscale US and confirmed by core needle biopsy was designated as the observation target. In the current study, pCR was defined as the absence of residual invasive cancer in both the breast and axillary lymph nodes, with cancer in situ permissible in some cases (ypT0/Tis ypN0).

Among all the study participants, the NAC regimen consisted of taxane-based, alkylator-based, or anthracycline-based chemotherapy, administered either as standalone treatments or in combination with neoadjuvant endocrine and/or anti-human epidermal growth factor receptor 2 targeted therapies, all selected based on the patients’ molecular subtypes.

Statistical analysis

All patients were divided into the pCR and non-pCR groups. Continuous variables are reported as mean ± standard deviation (SD), while categorical variables are presented as numbers (n) and percentages (%). The relationships between all independent variables and pCR were estimated using univariate logistic regression, and the crude odds ratios (OR), OR’s 95% Confidence Intervals (CI), and significance level were reported.

A multivariate logistic regression model employing the forward method and Wald test was used to identify the combination of independent variables with the best predictive power for achieving pCR. AUC analysis was used to confirm the diagnostic efficacy of continuous variables and the multivariate model for pCR, with AUCs for different variables tested for differences using DeLong’s test.

To evaluate the internal validity of the multivariate logistic regression model, bootstrap resampling was performed. Calibration plots and decision curve analysis (DCA) were also used to confirm the consistency between the probabilities estimated by the multivariable logistic regression model and the actual probabilities, as well as to assess the net benefit of applying this model to predict pCR. Finally, we provided a nomogram for this multivariate logistic regression model for practical clinical use.

All analyses were done using statistical software R (version 4.5.0). The two-tailed statistical significance level for all tests was set at P < 0.05.

Results

Patients’ baseline characteristics

In total, 107 patients with confirmed invasive BC were included in this study. The average age of the patients was 49.74 ± 11.28 years, with approximately half (45.8%) having reached menopausal status. The most common T stage was T2 (75.7%), and three-quarters of patients had lymph node metastasis. The histological type of the tumor was predominantly invasive breast carcinoma, non-specific type (IBC-NST, 103/107, 96.26%), with one case of Paget’s disease; the other types included two cases of invasive lobular carcinoma (ILC), and one case each of metaplastic carcinoma and medullary carcinoma. In 99 (92.5%) cases, the lesions had a Ki67 index ≥14%. There were 33 patients (30.8%) with positive echogenic rinds, and 76 patients (71.0%) exhibited expanded lesion scopes on mSMI compared with grayscale ultrasound (Table 1).

Table 1 Patient’s clinical characteristics.

Parameters	Mean ± SD or n (%)	
Age (year)	49.74 ± 11.28	
Menopausal status		
Yes	49 (45.8%)	
No	58 (54.2%)	
Tumor stage		
T1	2 (1.9%)	
T2	81 (75.7%)	
T3	10 (9.3%)	
T4	14 (13.1%)	
Lymph node metastasis		
Positive	81 (75.7%)	
Negative	26 (24.3%)	
Clinical stage		
II	54 (50.5%)	
III	53 (49.5%)	
ER status		
Positive	61 (57.0%)	
Negative	46 (43.0%)	
PR status		
Positive	49 (45.8%)	
Negative	58 (54.2%)	
HER2 status		
Positive	49 (45.8%)	
Negative	58 (54.2%)	
Ki-67		
≥ 14%	99 (92.5%)	
<14%	8 (7.5%)	
LD	3.88 ± 1.25	
SD	1.96 ± 0.72	
VI	13.45 ± 7.74	
Echogenic rinds		
Positive	33 (30.8%)	
Negative	74 (69.2%)	
mSMI Scope		
Expand	76 (71.0%)	
No expand	31 (29.0%)	
Notes.

Abbreviations ER estrogen receptor

PR progesterone receptor

HER2 human epidermal growth factor receptor 2

LD largest diameters

SD shortest diameters

VI vascular index

mSMI monochrome superb microvascular imaging

Comparison between pCR group and non-pCR group: clinicopathologic factors and multiparametric US parameters

The 107 patients were divided into two groups based on whether they achieved pCR: non-pCR group (n = 56) and pCR group (n = 51). As indicated, the pCR group had significantly higher negative rates of ER and PR (P < 0.001 and P = 0.005, respectively), and significantly higher positive rates of HER2 and echogenic rinds (P < 0.001 and P = 0.029, respectively). Patients in the pCR group also had significantly shorter LD and SD (P = 0.001 and P = 0.003, respectively) (Table 2).

Table 2 The grouped statistics and estimated crude OR of pre-NAC parameters to pCR.

Parameters	Non-pCR	pCR	Crude OR (95% CI)	P	
Sample size	56	51			
Age (year), mean ± SD	50.5 ± 11.7	48.9 ± 10.8	0.99 (0.96 to 1.02)	0.484	
Menopausal status, n (%)				0.802	
No	31 (55.4%)	27 (52.9%)	1		
Yes	25 (44.6%)	24 (47.1%)	1.10 (0.51 to 2.36)		
ER status, n (%)				<0.001	
Positive	41 (73.2%)	20 (39.2%)	1		
Negative	15 (26.8%)	31 (60.8%)	4.24 (1.87 to 9.58)		
PR status				0.005	
Positive	33 (58.9%)	16 (31.4%)	1		
Negative	23 (41.1%)	35 (68.6%)	3.14 (1.42 to 6.96)		
HER2 status				<0.001	
Positive	13 (23.2%)	36 (70.6%)	1		
Negative	43 (76.8%)	15 (29.4%)	0.13 (0.05 to 0.30)		
Tumor stage				0.205	
T1	1 (1.8%)	1 (2%)	1	–	
T2	38 (67.9%)	43 (84.3%)	1.13 (0.07 to 18.72)	0.931	
T3	6 (10.7%)	4 (7.8%)	0.67 (0.03 to 14.03)	0.794	
T4	11 (19.6%)	3 (5.9%)	0.27 (0.01 to 5.77)	0.404	
Lymph node metastasis				0.242	
Negative	11 (19.6%)	15 (29.4%)	1		
Positive	45 (80.4%)	36 (70.6%)	0.59 (0.24 to 1.43)		
Ki-67				0.200	
≥ 14%	50 (89.3%)	49 (96.1%)	1		
<14%	6 (10.7%)	2 (3.9%)	0.34 (0.07 to 1.77)		
Clinical stage				0.775	
III	27 (48.2%)	26 (51%)	1		
II	29 (51.8%)	25 (49%)	0.90 (0.42 to 1.91)		
mSMI Scope				0.602	
Expand	41 (73.2%)	35 (68.6%)	1		
No expand	15 (26.8%)	16 (31.4%)	1.25 (0.54 to 2.88)		
Echogenic rinds				0.029	
Negative	44 (78.6%)	30 (58.8%)	1		
Positive	12 (21.4%)	21 (41.2%)	2.57 (1.10 to 5.99)		
LD, mean ± SD	4.3 ± 1.3	3.5 ± 1.1	0.56 (0.40 to 0.80)	0.001	
SD, mean ± SD	2.2 ± 0.9	1.7 ± 0.4	0.32 (0.16 to 0.68)	0.003	
VI, mean ± SD	12.7 ± 7.3	14.3 ± 8.2	1.03 (0.98 to 1.08)	0.302	
Notes.

Abbreviations OR odds ratio

NAC neoadjuvant chemotherapy

pCR pathological complete response

SD standard deviation

n number

ER estrogen receptor

PR progesterone receptor

HER2 human epidermal growth factor receptor 2

mSMI monochrome superb microvascular imaging

LD largest diameters

SD shortest diameters

VI vascular index

The parameters following two cycles of NAC

After two cycles of NAC, patients who achieved pCR demonstrated a significantly higher proportion of lesions with no further expansion of the mSMI scope and disappearance of echogenic rind changes (P < 0.001 and P = 0.002). In terms of the rate of change at LD, SD, and VI, the patients in the pCR group demonstrated significantly higher values than those in the non-pCR group (all P < 0.001) (Table 3) (Fig. 1).

Table 3 The grouped statistics and estimated crude OR of parameters following two cycles of NAC to pCR.

Parameters	Non-pCR	pCR	Crude OR (95% CI)	P	
Sample size	56	51			
mSMI, n (%)				<0.001	
Haven’t expanded	3 (5.4%)	20 (39.2%)	1		
No change or expanded	53 (94.6%)	31 (60.8%)	0.09 (0.02 to 0.32)		
Echogenic rinds change, n (%)				0.002	
Disappear	4 (7.1%)	17 (33.3%)	1		
No change	52 (92.9%)	34 (66.7%)	6.50 (2.01 to 20.98)		
ΔLD (%), mean ± SD	18.0 ± 17.0	36.1 ± 21.7	1.05 (1.02 to 1.07)	<0.001	
ΔSD (%), mean ± SD	21.  ± 1.6	51.9 ± 14.4	1.12 (1.07 to 1.16)	<0.001	
ΔVI (%), mean ± SD	11.4 ± 43.9	61.3 ± 24.2	1.05 (1.03 to 1.07)	<0.001	
Notes.

Abbreviations OR odds ratio

NAC neoadjuvant chemotherapy

pCR pathological complete response

mSMI monochrome superb microvascular imaging

n number

SD standard deviation

LD largest diameters

SD shortest diameters

VI vascular index

The multivariate model in predicting pCR

Using a multivariate logistic regression model with forward selection and the Wald test, three factors were identified as having the strongest association with pCR: ΔVI (%), ΔSD (%), and SD. As shown in Table 4, a greater rate of change in VI and SD (Δ%) was associated with a higher likelihood of patients achieving pCR. In contrast, a more extended baseline SD was inversely associated with a reduced probability of achieving pCR. Although the AUC values of ΔVI (%) and ΔSD (%) were good (0.855 and 0.906, respectively), the combined multivariate model still exhibited the best AUC (0.963), which was significantly higher than that of any single factor (Table 5) (Fig. 2).

Figure 1 A 42 -year-old woman with triple-negative invasive breast cancer in the left breast. After receiving NAC, achieving pCR.

Before NAC: (A) Hypoechoic lesion with a diameter of 23 × 19 × 15 mm on grayscale, featuring an echogenic rind. (B) The lesion on the mSMI side has an expanded scope, compared displayed by grayscale ultrasound. (C) The vascular index was measured on the plane containing the most abundant vasculature, with a value of 6.5%. After two cycles of NAC: (D) The lesion has a diameter of 12 × 9 × 4 mm, with the rate of change at SD approximately 73.3%. (E) The lesion on the mSMI side has the same scope as that observed on grayscale ultrasound, with no additional expansion. (F) The vascular index was measured on the plane containing the most abundant vasculature, with a value of 2.0%. The rate of change at VI was approximately 69.2%. Note: Abbreviations: pCR, pathological complete response; NAC, neoadjuvant chemotherapy; mSMI, monochrome superb microvascular imaging; SD, shortest diameters; VI, vascular index.

The internal validation of the multivariate model

To evaluate the internal validity of the multivariate logistic regression model, bootstrap resampling (B = 1,000) was performed using the rms package in R. The results demonstrated strong performance in both discrimination and calibration.

The Dxy index (Somers’ Dxy) of the model was 0.927 in the original sample, indicating excellent discriminative ability. After bootstrap correction, Dxy was slightly reduced to 0.919 with a minimal optimism of 0.008. Similarly, the R2 index decreased modestly from 0.788 to 0.766, suggesting stable explanatory power.

Table 4 The multivariate model results of independent variables to pCR using forward auto-selection method and Wald test.

Parameters	Estimated B	S.E.	Wald	Adjusted OR (95% CI)	P	
ΔVI (%)	0.033	0.014	5.859	1.034 (1.006 to 1.062)	0.015	
ΔSD (%)	0.123	0.029	18.003	1.131 (1.068 to 1.197)	<0.001	
SD	−2.576	0.993	6.726	0.076 (0.011 to 0.533)	0.009	
Notes.

Abbreviations pCR pathological complete response

LD largest diameters

SD shortest diameters

VI vascular index

Table 5 The comparisons of AUCs between parameters following two cycles of NAC and multivariate model.

		P-value compared to the AUC of	
Parameters	AUC	ΔVI (%)	ΔSD (%)	SD	Multivariate model	
ΔVI (%)	0.855	–	0.228	0.004	<0.001	
ΔSD (%)	0.906	0.228	–	<0.001	0.012	
SD	0.666	0.004	<0.001	–	<0.001	
Multivariate model	0.963	<0.001	0.012	<0.001	–	
Notes.

Abbreviations AUC area under the receiver operating characteristic curve

NAC neoadjuvant chemotherapy

LD largest diameters

SD shortest diameters

VI vascular index

Figure 2 The ROC results of parameters following two cycles of NAC and multivariate model to pCR.

Note: Abbreviations: ROC, receiver operating characteristic; NAC, eoadjuvant chemotherapy; pCR, pathological complete response; SD, shortest diameters; ΔSD, the change rat io of shortest diameters; ΔVI, the change ratio of vascular index.

In Fig. 3, the original and corrected values for each parameter show similar results. In the calibration plot in Fig. 4, the distribution of the predicted and actual probabilities closely aligns with the 45-degree diagonal line. Both of these findings demonstrate the stability of the estimation.

Figure 3 The comparisons between original and corrected parameters after the internal validation of the multivariate logistic regression model using bootstrapping method (B = 1,000).

Figure 4 The calibration plot of the multivariate logistic regression model.

The decision curve analysis and nomogram

DCA was performed to assess the clinical utility of the prediction model (Fig. 5). The DCA showed that the model provided a higher net benefit than the “treat all” or “treat none” strategies across a range of threshold probabilities (0.4 to approximately 0.6), supporting the model’s potential value in predicting pCR. Figure 6 depicts the nomogram of this 3-factor multivariate logistic regression model for convenience in clinical practice.

Figure 5 The decision curve analysis plot of the multivariate logistic regression model.

Figure 6 The nomogram of the multivariate logistic regression model in prediction pCR.

Note: Abbreviations: pCR, pathological complete response; SD, shortest diameters; ΔSD, the change ratio of shortest diameters; ΔVI, the change ratio of vascular index.

Discussion

In the current study, we combined baseline SD with the rate of change in VI and SD (Δ%) to construct an easily accessible and cost-effective model for predicting pCR after NAC in BC patients. The model yielded favorable outcomes in terms of AUC, calibration, and DCA. This could guide the timely adjustment of treatment regimens for patients who do not achieve pCR, thereby increasing the rates of pCR and preventing toxic effects.

BC is a heterogeneous disease that is characterized by multiple molecular subtypes and genetic features. The treatment plan was primarily determined by molecular classification (Gradishar et al., 2024). Literature reports that ER-negative and/or HER2-positive status are more likely to achieve pCR (Chen et al., 2023a; Chen et al., 2024; Guo et al., 2024). Our study confirmed this as well: In the pCR group, 31 (60.8%) patients were ER-negative, and 36 (70.6%) were HER2-positive.

Currently, Chinese experts strongly recommend that individuals with BC undergoing NAC should receive routine ultrasound screenings every two cycles, primarily to monitor changes in the size of breast tumors (Shao et al., 2022). Our research revealed that, prior to NAC, both LD and SD in the non-pCR group were significantly larger than those in the pCR group, indicating statistically significant differences between the two groups. Previous studies (Liu, Tang & Chen, 2022) suggested that the larger the maximum diameter of the tumor before NAC, the lower the likelihood of achieving pCR. However, Xie et al. (2023) found that there was no statistically significant difference in the baseline lesion size, but there was a difference in the change rate of the LD and volume after two cycles of treatment. Moreover, the rate of volume change was an independent predictor of pCR. However, our research indicated that although the rates of change in both LD and SD were higher in the pCR group than in the non-pCR group after two cycles of treatment, ΔSD and SD were identified as having the strongest association with pCR. To the best of our knowledge, very few studies have reported the value of predicting pCR based on the SD of the lesion. Our research showed that ΔSD demonstrated outstanding performance, achieving an AUC value as high as 0.906. We hypothesized that because the breast is a flat organ, tumors tend to shrink in a parallel manner following NAC, irrespective of whether they undergo concentric or non-concentric shrinkage (Ballesio et al., 2017). In cases of non-concentric shrinkage, the primary lesion is divided into separate sublesions, which increases the measurement error for the LD (Park, Kim & Lee, 2022), whereas the SD is relatively easier to measure. Therefore, we considered SD and ΔSD as effective indicators for evaluating the response to NAC.

Echogenic rinds refer to a thick band of high echo rings surrounding the entire or part of a breast mass, which is a recognized malignant sign of breast tumors, with a positive predictive value as high as 88%–97.5% (Watanabe et al., 2021; Kim et al., 2023a). Some studies suggest that echogenic rinds are related to the displacement of surrounding tissues, edema, and infiltration of new microvessels or lymphatics, indicating that the tumor grows faster (Park et al., 2022). Tumors that proliferate rapidly are more likely to achieve pCR (Chung et al., 2022). Chung et al. (2022) found that triple-negative breast cancers with echogenic rinds before NAC were more likely to achieve pCR. Our results are consistent with these findings. A total of 33 lesions with echogenic rinds were identified, 21 of which achieved pCR. Among these, six patients had triple-negative breast cancer. Additionally, we conducted a longitudinal study to monitor the changes in echogenic rinds following two cycles of treatment. In the pCR group, echogenic rinds disappeared in 17 lesions. In contrast, in the non-pCR group, only four lesions showed disappearance, with no instances of regenerative echogenic rinds. Although there was a statistically significant difference between the two groups, this phenomenon was not confirmed to be an independent predictive factor. Cui et al. (2021) found that neither the presence nor absence of a pretreatment echogenic rind, nor the disappearance of the echogenic rinds after two cycles of treatment, was associated with pCR. However, Chen et al. (2023b) found that narrowing or disappearance of echogenic rinds was an independent predictor of pCR. However, the value of echogenic rinds in predicting the efficacy of NAC in BC remains controversial. Further research and discussions are required.

Tumor angiogenesis refers to the abnormal proliferation of blood vessels that penetrate cancerous tumors (Park & Seo, 2017). Clinical evaluation of tumor vascularity can assist in the diagnosis of breast cancer, determine a treatment strategy, and forecast the prognosis. Our research indicated that the VI was slightly elevated in the pCR group compared to the non-pCR group before NAC; however, this difference was not statistically significant. Qi et al. (2023) and Lee & Chang (2024) agreed with us, believing that breast cancer is a hypervascular heterogeneous mass, and the VI based on a two-dimensional (2D) view cannot comprehensively assess the entire tumor vascularity. Moreover, the vascularity of large tumors may be underestimated because the Doppler signal diminishes with depth. Study (Mo et al., 2023) suggest that NAC affects the angiogenic activity of tumors, potentially impeding their further development. Therefore, we believe that monitoring and analyzing changes in vascular characteristics could serve as biomarkers for evaluating the response to NAC. It was exciting to observe that after two cycles of NAC, the change rate of VI in the pCR group was significantly higher than that in the non-pCR group, demonstrating medium predictive potential. The AUC value was 0.855 (95% CI [0.785–0.926]). NAC can affect both the cells and stroma of the tumor, including its microvessels. Consequently, the response of the tumor vascular bed allows for indirect assessment of the tumor’s sensitivity to drug reactions (Pavlov et al., 2023). Therefore, the alteration rate of VI can serve as a valuable imaging biomarker for predicting pCR to NAC in breast cancer.

mSMI and cSMI are the two manifestations of SMI. However, compared with cSMI, reports on mSMI are relatively scarce. Diao et al. (2020) research found that the diagnostic performance of mSMI appears to be equivalent to that of CEUS. However, their concern was the difference in diagnostic efficacy between CEUS and mSMI in penetrating blood vessels or vascular features. We observed that in 76 lesions (71.03%), the lesion scope depicted in the mSMI images was larger than that shown in grayscale US. This characteristic also serves as a key indicator for CEUS in the diagnosis of breast cancer. We further investigated whether the scope continued to expand after two cycles of NAC treatment. The results indicated that 20 lesions did not show further expansion in the pCR group, whereas three lesions did not expand in the non-pCR group. We speculate that this may be due to the effect of chemotherapy drugs causing atrophy and collapse of the tumor neovasculature, which also reflects a reduction in tumor invasiveness. However, the mSMI0 and its changes were not included as indicators in the final predictive model. Further studies with a larger number of patients may help provide additional value to mSMI.

This study had several limitations. First, this was a single-center retrospective study with a small sample size. Second, while the molecular classification of breast cancer is closely related to the efficacy of NAC, we did not conduct a more in-depth stratified analysis based on the molecular subtypes. Third, the measurement of the quantitative parameter VI required scanning of the section with the most abundant blood flow. However, it was challenging to ensure that the same section was selected during the different treatment cycles. Finally, this study was fundamentally tied to two-dimensional ultrasound imaging. Future research should incorporate volumetric imaging of tumors, thereby providing better insights into microvascular changes with NAC.

Conclusion

In summary, the rate of change in SD following two cycles of NAC has emerged as a crucial dependent factor for predicting pCR. The nomogram constructed from the combined predictive models of ΔSD (%), SD, and ΔVI (%) demonstrated satisfactory performance, indicating that the nomogram is a reliable method for predicting pCR after NAC. However, further research using larger validation datasets is essential before this model can be practically applied.

Supplemental Information

Supplemental Information 1 Data

Supplemental Information 2 STROBE checklist

Abbreviations

US Ultrasound

SMI Superb microvascular imaging

cSMI Color superb microvascular imaging

mSMI Monochrome superb microvascular imaging

VI Vascular index

pCR Pathological complete response

NAC Neoadjuvant chemotherapy

BC breast cancer

AUC Area under the receiver operating characteristic curve

ER Estrogen receptor

PR Progesterone receptor

HER2 Human epidermal growth factor receptor 2

DCA Decision curve analysis

LD Largest diameters

SD Shortest diameters

MRI Magnetic resonance imaging

PPV Positive predictive value

BI-RADS Breast imaging reporting and data system

CEUS Contrast-enhanced ultrasound

IHC Immunochistochemistry

SISH Silver in situ hybridization

SD Standard deviation

n Number

% Percentage

OR Odds ratio

CI Confidence interval

IBC-NST Invasive breast carcinoma, non-specific type

ILC Invasive lobular carcinoma

2D Two-dimensional

Additional Information and Declarations

Competing Interests

Author Contributions

Human Ethics

Data Availability

The authors declare there are no competing interests.

Yanling Zuo conceived and designed the experiments, prepared figures and/or tables, authored or reviewed drafts of the article, and approved the final draft.

Yongtao Zhan conceived and designed the experiments, performed the experiments, prepared figures and/or tables, and approved the final draft.

Jie Zhou performed the experiments, prepared figures and/or tables, and approved the final draft.

Haoming Xia performed the experiments, prepared figures and/or tables, and approved the final draft.

Tao Li performed the experiments, prepared figures and/or tables, and approved the final draft.

Fan Zhou performed the experiments, analyzed the data, prepared figures and/or tables, and approved the final draft.

Chunyue Luo performed the experiments, prepared figures and/or tables, and approved the final draft.

Huafeng Zeng performed the experiments, prepared figures and/or tables, and approved the final draft.

Yingjia Li conceived and designed the experiments, analyzed the data, authored or reviewed drafts of the article, and approved the final draft.

The following information was supplied relating to ethical approvals (i.e., approving body and any reference numbers):

The study design and protocol were approved by the ethics committee of Affiliated Cancer Hospital & Institute of Guangzhou Medical University (No. KY-2025-08) date on February 5, 2025), and WRITTEN informed consent was obtained.

The following information was supplied regarding data availability:

The raw measurements are available in the Supplementary File.

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
