# Peer review of "Prediction of pathological complete response to neoadjuvant chemotherapy for invasive breast cancers based on longitudinal ultrasound and superb microvascular imaging: a single-center retrospective study"

_PeerJ, doi:10.7717/peerj.20171_

## Round 0.1 · original submission · Major Revisions

After careful consideration of the reviewers’ comments, I have decided that major revisions are required before your manuscript can be considered for publication. Please address all the points raised by both reviewers, with particular attention to the substantive concerns. I look forward to receiving your revised manuscript.

**Language Note:** The review process has identified that the English language must be improved. PeerJ can provide language editing services - please contact us at [email protected] for pricing (be sure to provide your manuscript number and title). Alternatively, you should make your own arrangements to improve the language quality and provide details in your response letter. – PeerJ Staff

Reviewer 1 ·

Basic reporting

This study retrospectively analyzed that whether the dynamic alterations in conventional ultrasound (US) and superb microvascular imaging (SMI) can act as predictors of pathological complete response (pCR) following neoadjuvant chemotherapy (NAC) in breast cancer (BC). Ultimately, by using multivariate logistic regression analysis, the authors combined baseline shortest diameters (SD) with the rate of change in vascular index (VI) and SD (Δ%) to construct an easily accessible and cost-effective model for predicting pCR after NAC in BC, which exhibited favorable outcomes in terms of AUC, calibration and decision curve analysis. Although there are still some limitations, the results of this study provide a new reliable method for predicting pCR after NAC in BC. Some comments may be addressed to improve the manuscript.
1.In the Methods section of Abstract, the period from January 2022 to December 2024 was the time of patients received NAC in the hospital, not the study period. Besides, it should be invasive BC (be consistent with the title), not advanced BC.
2.The abbreviations should be deleted for Keywords.
3.The Introduction section should be revised. In my opinion, the second and third paragraphs should be condensed to eliminate redundancy, with only retaining the components directly pertinent to the current study.
4.Two SMI modes, cSMI and mSMI, are introduced in the Introduction section, which one was be used for the patient in the current study. That need be made clear in the Materials and Method section.
5.“Short L0 and S0 (P=0.001 and P=0.003)” in the 3.2 section was wrong. It should be Short LD and SD. The author should check the whole manuscript carefully to avoid detailed errors.
6.The relevant description concerning Figure 1 need to be presented more detailed in the manuscript. Whether the patient described in Figure 1 belongs to the pCR group. This information needs to be added in the manuscript.
7.Figure 2 to Figure 6 need aesthetic improvement, particularly in font size and excessive white space.
8.Abbreviations should be defined in the text at first use. Moreover, if abbreviation used in Tables and Figures, the full name should be supplemented in Figure/Table legend. The author should check the whole manuscript.
9.The language of the manuscript still needs to be improved. Moreover, there are some errors concerning format. The authors should revise and check the whole manuscript carefully or sough help from a professional language polishing agency.

Experimental design

/

Validity of the findings

/

Reviewer 2 ·

Basic reporting

The paper titled "Prediction of Pathological Complete Response to Neoadjuvant Chemotherapy for Invasive Breast Cancers Based on Longitudinal Ultrasound and Superb Microvascular Imaging" is interesting. How to early predict the efficacy of neoadjuvant chemotherapy for breast cancer is a hot topic of clinical concern. However, there are still some problems that need to be solved.

Experimental design

1. The author should carefully check the full name of the abbreviation (including figure legends and graphs), the full English expression and grammar, etc. It is recommended that at least a professional proficient in English should proofread it. It will help the content of the article to be better understood.
2. The Figures could be more professional. For instance, the white space, resolution, font size and other aspects need to be further improved.

Validity of the findings

It is worth affirming that the author's indexing of references is very accurate, with a high proportion in the past three years.

---

## Round 0.2 · accepted · Accept

Both reviewers have carefully evaluated your revised manuscript and recommended acceptance. Based on their positive feedback and the improvements made, I am pleased to inform you that your manuscript has been accepted for publication.

Reviewer 1 ·

Basic reporting

The author's revision can address the initial doubts, and I agree to accept the revised manuscript

Experimental design

no comment

Validity of the findings

no comment

Reviewer 2 ·

Basic reporting

NO Comment

Experimental design

NO Comment

Validity of the findings

NO Comment

Additional comments

NO Comment